# An evaluation of the Ultimatum Game as a measure of irritability and anger

**Maria Gröndal** *, **Karl Ask**, **Stefan Winblad**

Department of Psychology, University of Gothenburg, Gothenburg, Sweden

* maria.grondal@psy.gu.se

## Abstract

The Ultimatum Game is an effective tool for understanding how social decision-making is influenced by emotions in both research and clinical settings. Previous findings have shown that the Ultimatum Game can evoke negative emotions, especially anger and aggression. In a sample of non-clinical adults ($N$ = 143) we evaluated the sensitivity of an anger-infused version of the Ultimatum Game to individual differences in anger and irritability. Findings showed significant relationships between anger and aggressive behaviors in the Ultimatum game, but no association between irritability and aggressive behavior were observed. This indicates that the anger-infused Ultimatum Game is a promising method for studying individual differences in trait anger and anger expression. However, the relationship between decision-making in the anger-infused Ultimatum Game and irritability is less straight forward and needs further investigation. Therefore, when studying the behavioral responses of irritability, it would be beneficial to capture other behaviors beyond aggressive responses.

## Introduction

Although early psychological models of decision-making mainly have focused on cognitive processes [1], contemporary models hold that emotions are powerful drivers of important decisions [1–6]. However, such emotional influences are not uniform and the mechanisms that shape behavioral responses to emotions are subject to an ongoing debate within emotion theories [7]. The Ultimatum Game (UG) can be an effective tool for understanding how social decision-making is influenced by emotions in both research and clinical settings [7, 8]. These games contain emotion-evoking elements of social interaction because fairness/injustice and reward/punishment concerns are influenced by the game structure and by the players' decisions. Therefore, the UG can inform researchers about tendencies of different clinical and non-clinical subgroups in the process of making decisions in emotionally charged social situations [8].

In clinical research, subgroups of individuals are often identified based on pathological conditions. However, many symptoms associated with pathological states are commonly experienced both across different psychiatric conditions and within the healthy population. One example of such symptoms is irritability [9–11]. However, although irritability is a characteristic of 15 psychiatric disorders according to the current edition of the Diagnostic and Statistical Manual of Mental Disorders [DSM-V; APA, 12], and is socially impairing in both intense and

**Data Availability Statement:** All data, analysis code, and research materials are available at https://osf.io/gnj85/

**Funding:** This research was supported by funding from the Swedish Association of

Neuropsychologists (Sveriges Neuropsykologers Förenings Stipendiefond).

**Competing interests:** The authors have declared that no competing interests exist.

milder forms [11, 13], the behavioral consequences of irritability are not yet clearly understood [11]. Therefore, the present study aimed to explore whether an anger-infused version of the UG, developed by Gilam et al. [14], can be used to measure behavioral expressions of individuals' self-reported irritability in a non-clinical population.

## The emotion-evoking aspects of the Ultimatum Game

The original version of the UG [15] presents a bargaining situation between two players. The first player (responder) decides whether to accept or reject offers from the second player (proposer) of how an amount of money should be split between them. Sometimes the offers are split fairly and sometimes unfairly to the proposer's favor. If the responder accepts the offer, both receive the proposed sum. If the offer is rejected, both players end up with nothing. According to standard economic theory [16] the rejection of an unfair offer is irrational. This theory posits that agents should prioritize the maximization of their material gains, disregard emotions, and remain indifferent to the well-being of others. Nevertheless, behavioral economics offers various theories that could elucidate why individuals might decline such offers in the UG [17]. In general, responders more or less always accept fair offers, but they often reject offers of less than 40% of the total amount [18, 19]. This pattern of rejections is only found when the responders believe they are playing against another human. If the responders believe they are playing against a computer, the rejection rate is significantly lower [20]. Thus, the responder's decisions are affected by the social interaction. Further, the responder's decisions are not driven only by financial self-interest, because they would receive a higher total sum if both the fair and unfair offers were accepted.

This economically irrational rejection behavior has been linked to emotional and social behavioral mechanisms. First, people can use rejection behavior to avoid injustice, meaning that individuals oppose unfair results, indicating their willingness to sacrifice some material gain to achieve fairer outcomes [17] and endorsing the punishment of those who seek to take advantage of others [21, 22]. Another theoretical framework suggests that receiving unfair offers in leader-follower games like the UG evokes negative emotions such as frustration and anger [23]. The game setup allows for the expression of negative emotions through elevated levels of punishing and aggressive behaviors, measured as increased rejection of offers [24, 25]. The important role of emotions in decision making in the UG is also supported by neurobiological evidence [26, 27]. For instance, heightened skin conductance activity [28] and changes in pupil size [29] has been associated with rejection of unfair offers, which indicates that emotional brain systems are involved when playing UG.

The emotional state which has been studied most extensively, and which is most strongly associated with punitive behavior in the UG, is anger [14, 24, 27, 30–32]. In previous studies, rejection rates in the UG has been used as a measure of aggression—the behavioral expression of anger [14, 33]. In the current study, an anger-infused version of the Ultimatum Game (AI-UG) was used, which was originally developed to empirically investigate anger [14].

## Anger and irritability

Anger is usually defined as an emotion that involves displeasure of varied intensity, from mild annoyance to intense fury [34], and is closely related to aggressive behavior [35, 36]. The causes of anger are characterized by the experience of frustration (e.g., goal hindrance) or being personally insulted [e.g., treated unfairly, blamed or unjustified; 35]. Anger proneness can be seen as a consistent personality trait, temperament, or characteristic [37, 38], and is related to impaired social and mental wellbeing [39]. However, anger can also be understood as a multi-faceted psychobiological phenomenon which can be expressed and experienced in

multiple ways [40, 41]. In the current study, the Spielberger State-Trait Anger Expression Inventory 2 [STAXI-2; 34] was used to measure both the state and the trait components of anger and to distinguish between inward and outward expressions and control of anger. State anger refers to the current experience of angry feelings. Individuals with high trait anger have a general tendency to interpret many situations as annoying or frustrating and, to a high extent, react to the situations with elevated anger and aggression [34]. Further, individuals with higher trait anger have been found to exhibit more aggressive behavior across a number of domains [42], including rejection behavior in the UG [43].

In the scientific literature, irritability is generally considered to be closely related to anger [44, 45]. Irritability is broadly defined as excessive sensitivity to sensory stimuli and a lowered threshold for responding to the stimuli with anger or aggressive behavior [10, 11, 46–49]. Further, irritability is also commonly referred to as a personality dimension. More precisely, it represents a dimension of one's personality marked by a predisposition to become angry and respond strongly to minor provocations and disagreements [50]. This definition sets it apart from the full-blown emotional state of anger as well as from aggression, which refers to the behavioral expression of anger.

Based on this definition of irritability, it could be assumed that individuals with high irritability tend to become angrier and to display more aggressive responding when playing the AI-UG compared with individuals with low irritability. In addition to the established assumption that choices or the outcomes of choices can generate emotional reactions, it has also been shown that one's current emotional state or mood influences the process of decision-making [4, 51]. Thus, because participants with high (vs. low) levels of irritability are likely to perceive unfair offers as more anger-provoking, it is expected that they will display a higher tendency to respond with aggression, which in the AI-UG is expressed in higher rejection rates.

## The current study

The overall aim of the current study was to increase the general understanding of the construct of irritability and provide clues to yet unexplained parts of its definition. There has been limited research on the behavioral outcomes associated with irritability. Consequently, the present study aims to explore how the experience of irritability influences social decision-making. The currently existing definitions of irritability suggest that it may be experienced predominantly within the individual without explicit expressions, but at the same time it is defined as a lowered threshold for experiencing anger and aggression [9, 52]. These somewhat contradictory definitions will be further investigated in the present study.

The specific goal was to evaluate the sensitivity of the AI-UG to individual differences in irritability in terms of behavioral (accepting/rejecting offers) and emotional (self-reported) measures. Based on this aim, we formulated the following hypotheses:

H1. Participants with higher self-reported irritability will display more aggressive behavior, as measured by (a) higher rejection of medium fairness offers (in which participant receive 40% or 30% of the total sum), (b) higher rejection of unfair offers (in which participant receive 20% or 10% of the total sum), and, consequently, (c) lower total gain compared with participants with lower trait irritability.

H2. Participants with higher self-reported irritability will report a larger pre–post game increase in anger compared with participants with lower irritability.

As the relationship and differences between the constructs irritability and anger have not been clearly established, we also wanted to examine the overlap between the two constructs. Therefore, beyond the above confirmatory aims of the study, we explored the relationships

between behavioral measures of the AI-UG and self-reported measures of irritability and trait anger and expression and control of anger.

The research plan and hypotheses for the study were preregistered on the Open Science Framework (OSF) and are available here [https://osf.io/gnj85/].

## Materials and methods

### Participants and statistical power

The study included 143 individuals with ages ranging between 18 and 71 years ($M$ = 33.1), of which 103 (72.0%) were women, 39 (27.2%) were men, and 1 (0.7%) identified with "other" gender. In terms of current occupation, 114 participants (79.7%) reported they were employees or students, 13 (9.1%) were on parental or medical leave, 6 (4.2%) were retired, and 3 (2.1%) "other". Participants had an average of $M$ = 15.58 years of education ($SD$ = 3.24, $Mdn$ = 15). The participants were recruited from two different voluntary research participant pools and through announcements at a public library in [Blinded for peer review]. In the screening phase, participants filled out an online survey of emotional measurement scales, which included screening for irritability levels [Brief Irritability Test, BITe; 10] and an anger instrument [State-Trait Anger Expression Inventory 2, STAXI-2; 34]. The participants were also asked to indicate whether they agreed to being contacted again for a second study. When responses from 400 participants had been collected in the screening phase, individuals with the highest and lowest scores, respectively, on the irritability measure were invited to the follow up study. We used a quartile split to identify individuals who rated themselves as the least (bottom quartile) and the most (top quartile) irritable [53]. A total of 225 participants were asked to participate in the second study, out of which 82 with low irritability and 61 with high irritability participated (for a non-response analysis, see supplemental material on OSF [https://osf.io/gnj85/]). In total, 146 participants took part in the second study, but three participants were discarded from analyses: two due to technical issues during the lab study, and one due to having participated in another study using a similar game task. Further information about the non-response participants in the second study are available on OSF [https://osf.io/gnj85/].

Prior to the data collection, a sensitivity analysis was performed using the G*Power 3.1 [54], based on a two tailed α of .05, desired statistical power of .80, and a sample size of 200 participants. The analysis showed that we would be able to detect an effect size of $r$ = .19. When calculating the power based on the final sample size of 143 participants, the ability to detect an effect size of $r$ = .23 at 80% power was found.

### Instruments

In this section, the instruments that were analyzed for the current study are presented. The full questionnaire can be found at [https://osf.io/gnj85/]. An impulsivity measure (UPPS Impulsive Behavior Scale) was also included in the pre-registration but is not part of the current study. Additionally, an article on affective responding and impact of the COVID-19 pandemic has already been published within this research project (based on the self-reported data collected in the online pre-screening survey) and can be found at [Blinded for review].

**The Brief Irritability Test (BITe).**   The Brief Irritability Test [BITe; 10] was used to measure irritability. Participants rated the frequency with which they had experienced each of five symptoms of irritability in the last two weeks (e.g., "I have been grumpy"), using a six-point scale (1 = *never*, 6 = *always*). Participants filled out the BITe scale twice, first, as part of an online questionnaire (screening) and, second, as part of an experimental session. The total BITe score was calculated for the analyses by summing the ratings of all items, with higher scores indicating higher irritability. A Swedish version of the scale was used in this study,

translated by the authors in collaboration with two bilingual, native English speakers. The BITe scale displayed strong internal reliability both in the first (screening) session (Cronbach's $\alpha$ = .94, $\omega_{total}$ = .95) and in the second session ($\alpha$ = .94, $\omega_{total}$ = .95).

**State-Trait Anger Expression Inventory (STAXI-2).** The State-Trait Anger Expression Inventory 2 [STAXI-2; 34] was used to measure trait anger as well as anger expression and control, using four-point rating scales (1 = *not at all/almost never* to 4 = *very much/almost always*). The *trait anger* subscale measures the tendency to experience and express anger in different situations and consists of 10 items (e.g., "I am quick tempered"; ordinal $\alpha$ = .89, ordinal $\omega_{total}$ = .86). Ordinal indices of reliability are reported for measures using rating scales with five or fewer scale points [55]. Four subscales measure to the individual's tendency to express and control anger, each consisting of eight items. *Anger expression-out* (AX-O; e.g., "I express my anger") measures how often anger is expressed towards other persons and objects (ordinal $\alpha$ = .76, ordinal $\omega_{total}$ = .67). *Anger expression-in* (AX-I; e.g., "I keep things in") measures how often angry feelings are suppressed (ordinal $\alpha$ = .81, ordinal $\omega_{total}$ = .79). *Anger control-out* (AC-O; e.g., "I control my temper") measures how often angry feelings are being inhibited from outward expressions (ordinal $\alpha$ = .76, ordinal $\omega_{total}$ = .67). A*nger control-in* (AC-I; e.g., "I take a deep breath and relax") measures how often angry feelings are controlled by calming down (ordinal $\alpha$ = .89, ordinal $\omega_{total}$ = .88). For each subscale, the respective items were summed to form a composite score. Lastly, *anger expression index* (AX index) measures total anger expression and was calculated using the following formula: AX-O + AX-I–(AC-O + AC-I) + 48. For the present study, we used the Swedish translation developed by Lindqvist et al. [56]. Participants filled out the STAXI-2 only during the online screening.

**The Anger-Infused Ultimatum Game (AI-UG).** An anger-infused version of the Ultimatum Game (AI-UG), containing brief (sometimes provocative) interpersonal messages and developed by Gilam et al. [14], was used in this study. In the game, participants were asked to decide whether to accept or reject other players' (proposers) offers of how an amount of money should be split between them. Participants were informed that if they rejected an offer, neither they nor the proposer would receive any money for that turn. The game included 30 different offers, one from each of 30 purported proposers, together with an interpersonal message. Each offer involved the allocation of 100 SEK (approximately 10 USD). Participants were informed that they would be playing for real money and would be rewarded with a percentage of their earnings in the game in cash afterwards. To increase participants' motivation, they were also told that the percentage was contingent on how much they earned; the more they earned, the higher the percentage they would get by the end of the game. In reality, all participants received a payment equivalent to 10% of the total amount earned during the game. The offer types were defined as fair (50/50), medium unfair (40/60, 30/70), and unfair (20/80, 10/90), exclusively with the greater profit to the proposer. The order of the presented offers was randomized across the participants. The interpersonal messages were congruent with the fairness of the offers. The fair offers were presented together with a non-aggressive message (e.g., "I give as much as I take"), while unfair offers were presented together with an offensive message (e.g., "come on, loser!!!"). The offensiveness of the messages increased with the unfairness of the offers. In line with Gilam et al [14], the level of anger elicited by the interpersonal messages was validated in an pilot survey online. (For further details see the supplementary material on OSF [https://osf.io/gnj85/]).

**Emotion ratings.** Before and after playing the AI-UG in the experimental session, participants were asked to rate distinct state emotions using items from the Discrete Emotions Questionnaire [DEQ; 57] Eight state emotions, represented by two items each, were included: anger (anger, mad), disgust (grossed-out, revulsion), fear (scared, fear), anxiety (nervous, worry), sadness (lonely, sad), desire (longing, wanting), relaxation (calm, relaxation), and happiness

(liking/happy, enjoyment). Participants were asked to indicate to what extent they currently experienced each emotion on a scale from 1 (not at all) to 7 (an extreme amount). The ratings for the two items representing anger were combined into composite measures for the pre-game ratings ($\alpha$ = .87, $\omega_{total}$ = .89) and the post-game ratings ($\alpha$ = .78, $\omega_{total}$ = .83), respectively. Since we were interested in the change in state anger as a result of playing the AI-UG, an anger change score was calculated by subtracting the pre-game rating from the post-game rating.

## Procedure

In the online screening procedure, participants filled out a questionnaire including several self-report instruments presented in randomized order. The instruments used in the current study were BITe and STAXI-2. Participants provided informed consent before filling out the questionnaire. The screening procedure took place between June 19 and September 1, 2020.

The second session of the study took place either in the lab at the Department of [Blinded for review], University of [Blinded for review] ($n$ = 73), or online ($n$ = 70). The original research plan was to exclusively collect data in the lab; however, restrictions due to the COVID-19 pandemic complicated the data collection. Therefore, approximately halfway into the data collection, we decided to adapt the lab study procedure to an online-friendly version and continue collecting data online. The original lab study included two games; the AI-UG and the Point Subtraction Aggression Paradigm [PSAP; 58]. However, the PSAP could not be adapted to the online procedure and was therefore dropped.

Participants in the lab version of the study were greeted by the experimenter, received oral instructions about the testing procedure, and then filled out the BITe and the emotion ratings on a computer. The emotion ratings were filled out before and after each of the two games, thus four times in total. To control for any order effects, the order of the games was randomized, and the two games were separated by a neutral filler task. During the filler task, participants were instructed to locate and count all instances of the letter 'a' in a brief, neutral text, marking each one with a pencil. They were informed that the task assessed accuracy and concentration, and that the time to complete the task was not being recorded. Participants in the online version of the study first joined a video meeting online where an experimenter gave oral instructions for the study. To mimic the lab version as much as possible, participants were told to sit undisturbed in front of a computer during the entire session. Directly after the instructions, participants received a link to the online survey. As in the lab version, online participants first completed the BITe and emotion ratings before playing the AI-UG. After the game, participants completed the emotion ratings a second time. The second session of the study took place between August 21 and December 21, 2020.

The research was carried out in accordance with the guidelines for good research practice issued by the Swedish Research Council [59]. The Swedish Act Concerning the Ethical Review of Research Involving Humans [SFS 2008:192] regulates the types of research involving humans that shall undergo ethics review. Because the current research did not fulfil any of the conditions that necessitate review [The Swedish Research Council, 2017, p 30], it did not undergo formal ethics review. Nonetheless, the study was carefully planned in collaboration with a representative of the Swedish Ethical Review Authority to conform with internationally accepted standards for research ethics [60].

## Data analyses

The significance level was set at 5% and all computations were performed in R (version 4.1.2).

Before testing our hypotheses, we examined the intra-individual differences between the online screening (BITe1) and the second session (BITe2) with independent $t$-test. As the data

showed large intra-individual differences between the online screening (BITe1) and the second session (BITe2), both measurement occasions were included in the analyses.

To exclude the possibility that the focal relationships between emotional measures and behavioral responses in AI-UG differ as a function of whether participants took part in study in the lab or online, interaction terms were tested in multiple linear regression models with the emotional measures as focal predictor variables, group (lab or online) as proposed moderator, and rejection rates and total gain in the AI-UG as the outcome variables.

To test our first hypothesis (H1), bivariate correlations were calculated between the BITe irritability measures (BITe 1 and BITe 2) and the behavioral measures in the AI-UG (total gain and rejection of medium and unfair offers, respectively). Because of the relatively large within-participant differences between the BITe 1 and BITe 2 scores, we also explored whether participants with more consistent irritability levels displayed stronger correlations between irritability and AI-UG behaviors. We examined this potential interaction in hierarchical linear regression models, where the BITe2 score and the absolute change between BITe1 and BITe2 were entered as predictors in Step 1 and AI-UG behaviors served as outcome measures. In Step 2, an interaction term combining the BITe2 score and the absolute change score was added to the model.

To examine our second hypothesis (H2) bivariate correlations were calculated between irritability (BITe 1) and change in experienced anger as a result of playing the AI-UG. For exploratory purposes, a corresponding correlation was computed based on the more recent rating of irritability (BITe 2).

Finally, because the hypothesis-testing analyses only examined pairwise correlations between irritability and AI-UG outcomes, a series of exploratory hierarchical regression models were conducted to explore the unique contributions to AI-UG outcomes from irritability, trait anger, and anger expression and control. In these models, the AI-UG sum score served as the outcome measure, BITe 1 was entered as a predictor in a first step, the STAXI trait subscale in a second step, and the STAXI anger expression and control subscales in the third step.

None of the reported regression models exhibited signs of problematic multicollinearity (VIF $\leq$ 2.55, Tolerance $\geq$ 0.39).

All data, analysis code, and research materials are available at [https://osf.io/gnj85/].

## Results

### Preliminary analyses

The average BITe scores measured in the online screening (BITe1; $M$ = 13.57) and in the second session (BITe2; $M$ = 12.94) did not differ significantly from each other, $t(142)$ = 1.482, $p$ = .140, $d$ = .118, 95% CI [-0.11, 0.35]. However, considerable intra-individual changes between the first and second BITe ratings were observed, as illustrated in Fig 1. A significant positive relationship were observed between the first and second BITe ratings, $r(141)$ = .563, $p < .001$, 95% CI [0.44, 0.67].

Multiple regression models did not detect any significant interactions between the focal emotional measures and means of participation (lab or online) on rejection rates and total gain in the AI-UG. Therefore, means of participation will be disregarded in the following analyses. For further details on the analyses investigating the role of lab vs. online participation, see supplement on OSF [https://osf.io/gnj85/].

### Relationships between irritability and acceptance rates in the AI-UG

No correlation between BITe 1 irritability and the behavioral measures in the AI-UG were observed, thus our first hypothesis (H1) was not supported, see Table 1. Similarly, there were

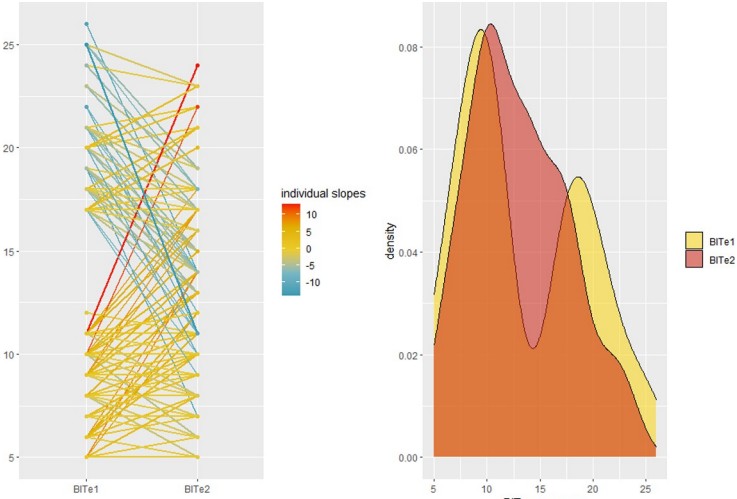

**Fig 1.** Individual Slopes (left panel) and Sample Distributions (right panel) of BITe Scores Measured at Time 1 and Time 2. *Note.* One participant was removed from the figure of individual slopes (left panel) due to an extreme discrepancy between measurement occasions, offsetting the line-color gradient.

no significant correlations between BITe 2 irritability and the AI-UG measures. Further, we found no significant association in the exploratory hierarchical regression model between the consistency of participants' irritability level over time and AI-UG outcomes. For further details on these analyses, see supplement on OSF [https://osf.io/gnj85/].

**Table 1. Correlations between measures of irritability and anger and total gain and offer rejections in the AI-UG (N = 143).**

| | *M* | *SD* | BITe 1 | BITe 2 | AI-UG total gain | AI-UG rejections (medium) | AI-UG rejections (unfair) |
|---|---|---|---|---|---|---|---|
| BITe 1 | 13.45 | 5.97 | - | **.563** [.176, .470] | -.136 [-.293, .029] | .139 [-.025, .297] | .077 [-.088, .238] |
| BITe 2 | 12.91 | 4.60 | **.563** [.176, .470] | - | -.103 [-.263, .061] | .087 [-.079, .248] | .087 [-.077, .249] |
| STAXI trait | 15.23 | 4.41 | **.571** [.449, .672] | **.437** [.294, .561] | **-.292** [-.435, -.134] | **.271** [.112, .417] | **.238** [-.076, .386] |
| STAXI AC-I | 22.19 | 5.46 | -.151 [-.308, .013] | .004 [-.160, .167] | -.068 [-.229, .097] | .077 [-.088, .238] | .051 [-.114, .213] |
| STAXI AC-O | 25.62 | 3.71 | **-.379** [-.511, -.229] | **-.251** [-.399, -.091] | .029 [-.135, .192] | -.006 [-.170, .158] | -.036 [-.198, .128] |
| STAXI AX-I | 17.81 | 4.76 | **.470** [-.331, 589] | **.382** [.232, .514] | **-.220** [-.374, .062] | **.231** [.700, .381] | .159 [-.005, .314] |
| STAXI AX-O | 12.13 | 2.94 | **.378** [.228, .511] | **.258** [.098, .405] | **-.285** [-.429, -.126] | **.266** [.106, .412] | **.227** [.065, .376] |
| STAXI AX index | 30.13 | 10.53 | **.529** [.400, .638] | **.330** [.176, .470] | -.156 [-.312, .008] | .141 [-.023, .298] | .122 [-.043, .280] |
| *M* | | | - | - | 567.48 | 188.74 | 141.33 |
| *SD* | | | - | - | 197.30 | 146.87 | 63.67 |

Note. Correlation coefficients are displayed with 95% CIs. Significant correlations are displayed in boldface. AI-UG = Anger-infused Ultimatum Game, BITe = Brief Irritability Test; STAXI trait = STAXI anger trait subscale; STAXI AC-I = STAXI anger control-in subscale; STAXI AC-O = STAXI anger control-out; STAXI AX-I = STAXI anger expression-in; STAXI AX-O = STAXI anger expression-out; STAXI AX index = STAXI anger expression index.

### Relationships between irritability and pre-game–post-game state anger change

Descriptive statistics and effect sizes of all emotion-state ratings before and after playing the AI-UG are presented in Table 2. Failing to support our hypothesis (H2), no correlation between BITe 1 irritability and change in anger state was found, $r(141) = -.117$, $p = .164$, 95% CI [-0.28, 0.05]. However, a negative relationship between BITe 2 irritability and change in state anger was found, $r(141) = -.174$, $p = .036$, 95% CI [-0.33, -0.01]. This finding indicates that participants who reported higher levels of irritability immediately before playing AI-UG reported a lower increase in state anger between their pre-game and post-game ratings.

### Exploratory analyses

In addition to the confirmatory analyses above, some additional patterns were observed in the correlational analyses reported in Table 1. First, significant relationships between the AI-UG behavioral measures and the trait anger subscale were observed, indicating that participants with higher trait anger tended to reject more offers and receive a lower total gain. Second, similar associations were observed between the AI-UG behaviors and the anger subscales AX-O and AX-I. These findings indicate that participants with a stronger tendency to express anger outwardly or suppress angry feelings rejected more offers, thus earning less, when playing the AI-UG.

The results from the exploratory hierarchical models (see Table 3) showed that in Step 1, when only BITe1 was included, no significant association with the AI-UG sum score was found. In the second step, when trait anger was added, a significant improvement from the first model was observed. No model improvement was observed when the expression and control subscales of the STAXI-2 were included in Step 3. In sum, confirming the bivariate correlations reported in Table 1, the models indicate that trait anger, but not irritability, is significantly associated with the sum score in the AI-UG. Furthermore, individual differences in anger expression and control did not explain significant portions of variance above that which is accounted for by trait anger.

## Discussion

The present study aimed to further the developing literature on irritability and social decision-making in a non-clinical setting. In our sample, the results did not show any significant relationships between self-reported irritability and rejection behavior in the AI-UG; thus, the first hypothesis of the current study was not supported. The second hypothesis also failed to receive support as no significant relation was observed between irritability measured during the

**Table 2. Descriptive statistics and Cohen's d of the emotion-state ratings before and after playing the AI-UG.**

| Emotion state | Pre | | Post | | | |
|---|---|---|---|---|---|---|
| | *M* | *SD* | *M* | *SD* | *d* | *95% CI* |
| Anger | 3.03 | 1.86 | 3.11 | 1.72 | .042 | [-.276, .190] |
| Disgust | 2.45 | 1.17 | 2.62 | 1.38 | .131 | [-.369, .102] |
| Fear | 2.87 | 1.69 | 2.38 | 1.01 | **-.357** | [.122, .591] |
| Anxiety | 4.22 | 2.55 | 3.19 | 1.79 | **-.469** | [.233, .705] |
| Sadness | 4.09 | 2.66 | 3.36 | 2.05 | **-.309** | [.075. .543] |
| Desire | 5.81 | 2.94 | 4.66 | 2.46 | **-.426** | [.191, .661] |
| Relaxation | 8.47 | 2.64 | 8.24 | 2.68 | -.087 | [-.146, .320] |
| Happiness | 7.98 | 2.52 | 7.68 | 2.63 | .117 | [-.116, .350] |

**Table 3. Exploratory hierarchical regression results for AI-UG sum score.**

| Variable | *B* | 95% CI for *B* | | *SE B* | β | *R²* | Δ*R²* |
|---|---|---|---|---|---|---|---|
| | | *LL* | *UL* | | | | |
| **Step 1** | | | | | | .018 | .018 |
| BITe 1 | -4.49 | -9.96 | 0.97 | 2.76 | -.14 | | |
| **Step 2** | | | | | | .087 | .069** |
| Constant | 763.25 | 649.58 | 876.91 | 57.49 | | | |
| BITe 1 | 1.52 | -4.92 | 7.97 | 3.23 | .05 | | |
| STAXI trait | -14.16** | -22.81 | -5.51 | 4.38 | -.32 | | |
| **Step 3** | | | | | | .126 | .039 |
| BITe 1 | 2.09 | -4.93 | 9.10 | 3.55 | .06 | | |
| STAXI trait | -9.12 | -20.38 | 2.14 | 5.69 | -.21 | | |
| STAXI AC-I | -0.89 | -8.69 | 6.92 | 3.95 | -.02 | | |
| STAXI AC-O | -4.89 | -17.49 | 7.71 | 6.37 | -.09 | | |
| STAXI AX-I | -4.71 | -12.92 | 3.49 | 4.15 | -.11 | | |
| STAXI AX-O | -11.86 | -26.38 | 2.65 | 7.34 | -.18 | | |

*Note.* ** *p* < .01

screening procedure and pre-game–post-game changes in state anger. Unexpectedly, higher irritability measured in the second session was associated with a lower increase in state anger when playing the AI-UG, which was opposite to the predicted direction. Significant relationships between decision-making in the AI-UG and trait anger were detected, however, such that participants with higher trait anger tended to reject more offers and receive a lower total gain.

The failure to find the predicted relationships between irritability and AI-UG rejection rates may be related to the notion that irritability is a condition associated primarily with inward experience rather than outward expression [9, 10]. Thus, the externally oriented behavioral consequences of irritability may not be pronounced or consistent enough to be reliably detected by the AI-UG. Another possible explanation, based on theories defining irritability as a less extreme form of anger [61], could be that low frustration and anger (as in irritability) isn't enough to induce the aggressive behavior that the AI-UG can capture. Experiences of irritability has also been related to lower levels of arousal and tension than what usually occurs with anger [62]. Trait anger, on the other hand, is more strongly associated with outward aggressive expressions [34] and punishing behaviors [14], which may explain its relationship with AI-UG performance. Similarly, the anger expression subscales were positively correlated with rejection rates in the AI-UG. Not surprisingly, participants reporting a stronger tendency to express anger outwardly tended to reject more offers. Less expected, however, was the finding that participants with a stronger tendency to suppress angry feelings also tended to reject more offers. One possible explanation of this exploratory finding is that individuals scoring high on the AX-I subscale struggle primarily with the suppression of socially disapproved physical and verbal expressions of anger. Because the rejection of offers in the AI-UG does not constitute either form of aggression, and may not even be perceived as socially undesirable, it may not be suppressed to the same extent as other forms. Importantly, however, the exploratory hierarchical regression analysis revealed that the anger expression subscales did not uniquely explain variation in the AI-UG outcome when irritability and trait anger had already been included in the model.

Beyond the potential conceptual explanations above for the general lack of behavioral correlates of irritability, it is possible that the variation in participants' experienced irritability was

not large enough for the AI-UG performance measures to capture any behavioral differences. The behavioral consequences of irritability may be more prominent in patients with clinically significant levels of irritability whereas behavioral variability may be low in non-clinical populations.

Beyond the quantitative differences between normal and pathological irritability in levels of suffering, dysfunction, duration, and intensity, it has not yet been established whether irritability differs qualitatively between these two groups among adults [10, 13]. When behavioral responses of pathological irritability have been studied in children and adolescents, they have been described as deviating from behavioral norms [50]. Additional studies of behavioral responses in adults, including both pathological and normal irritability, should be conducted to ascertain any behavioral differences between the groups. Importantly, since the current results showed that AI-UG rejection rates were correlated with trait anger, but not with irritability, the question of differences in behavioral responses between anger and irritability needs to be further investigated.

The repeated measurements of BITe indicated considerable intra-individual changes from the first to the second measurement. The BITe scale asks about subjectively experienced irritability during the last two weeks, and the current findings indicate that irritability as captured by BITe can fluctuate considerably over time. Previous definitions of irritability hold that it can emerge as a result of biological deficits, such as lack of sleep, hunger, pain, and hormonal fluctuations [11]. Given that such factors vary within the individual over time, corresponding changes in irritability would be expected. The developers of BITe have argued that the scale likely captures a combination of trait and state components [10], and it remains for researchers to establish the relative contribution of each component to the total score. To investigate whether the instability of irritability scores could explain the lack of a relationship between irritability and AI-UG behaviors, we explored if the relationship was stronger among participants with more consistent irritability levels. The analyses failed to support this possibility.

Another possible explanation for the low test-retest reliability of the BITe can be derived from the sampling procedure used in the current study. We used the Extreme Groups Approach [EGA; 53] to select participants with extreme scores in the irritability ratings at the screening stage. While the EGA has the potential advantages of increasing statistical power and lowering financial costs, one general disadvantage is that the approach assumes that extreme scores in a sample represent true scores in the population. However, extreme cases in one measurement tend not to remain as extreme when measured again at a later time, a phenomenon known as regression towards the mean [63]. This methodological issue could be partially addressed in future studies by carefully selecting scales of high reliability and collecting repeated measures before selecting groups using the EGA [53].

## Limitations

A few limitations of the present study should be acknowledged. Firstly, the BITe's nature as a measure of experienced irritability over the past two weeks yields it a measure of symptoms rather than capturing the trait dimension of irritability. However, the choice of the BITe measure was deliberate, as it incorporates items that primarily gauge frustration at a lower threshold and not outwardly behavioral outcomes of the experienced frustration, as opposed to measuring anger and aggression (11). Whether the timeframe of the BITe measure or its specific focus on irritability (as distinct from anger and aggressiveness) is the primary reason for the null findings in our study remains uncertain. Nevertheless, we recommend that future research employs an irritability measure that more directly assesses the trait dimension to explore whether the AI-UG can capture individual variations in irritability. Second, the

current research was designed to investigate general correlational patterns, as opposed to studying distinct patterns in different subpopulations. Thus, we did not (a) request participant information on variables that may be perceived as sensitive in the Swedish cultural context (e.g., ethnicity, socioeconomic status, clinical diagnoses) or (b) collect a sample large enough to perform informative subgroup analyses. As a consequence, we are unable to draw conclusions about the stability of our findings across various demographic characteristics or the extent to which they apply to both clinical and non-clinical populations.

## Conclusion

Taken together, the AI-UG provides a promising platform for studying individual differences in trait anger and anger expression. However, the relationship between irritability and decision-making in the AI-UG is less straightforward and needs to be further investigated. The current findings suggest that irritability, compared with anger, is less strongly associated with aggressive behavior. Therefore, when studying the behavioral responses of irritability, it would be beneficial to capture other behaviors beyond aggressive responses. Finally, because the failure to detect a relationship between irritability and decision-making in the AI-UG may have been caused by limited variability in participants' irritability scores, we suggest that future similar studies also include participants with pathological levels of irritability on a trait level.

## Supporting information

**S1 File.**
(DOCX)

## Acknowledgments

The authors would like to thank Malin Ekelund, Linnea Koponen, and Erik Loberg for their assistance with data collection.

## Author Contributions

**Conceptualization:** Maria Gröndal, Karl Ask, Stefan Winblad.

**Formal analysis:** Maria Gröndal.

**Investigation:** Maria Gröndal.

**Methodology:** Maria Gröndal, Karl Ask, Stefan Winblad.

**Project administration:** Maria Gröndal.

**Software:** Maria Gröndal.

**Supervision:** Karl Ask, Stefan Winblad.

**Visualization:** Maria Gröndal.

**Writing – original draft:** Maria Gröndal.

**Writing – review & editing:** Karl Ask, Stefan Winblad.

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
