## [Decision Letter · Decision Letter 0]

2 Jan 2024

PONE-D-23-32978An Evaluation of the Ultimatum Game as a Measure of Irritability and AngerPLOS ONE

Dear Dr. Gröndal,

Thank you for submitting your manuscript to PLOS ONE. After careful consideration, we feel that it has merit but does not fully meet PLOS ONE’s publication criteria as it currently stands. Therefore, we invite you to submit a revised version of the manuscript that addresses the points raised during the review process.

We look forward to receiving your revised manuscript.

Kind regards,

Junchen Shang

Academic Editor

PLOS ONE

2. In the online submission form, you indicated that your data will be submitted to a repository upon acceptance.  We strongly recommend all authors deposit their data before acceptance, as the process can be lengthy and hold up publication timelines. Please note that, though access restrictions are acceptable now, your entire minimal  dataset will need to be made freely accessible if your manuscript is accepted for publication. This policy applies to all data except where public deposition would breach compliance with the protocol approved by your research ethics board. If you are unable to adhere to our open data policy, please kindly revise your statement to explain your reasoning and we will seek the editor's input on an exemption.

Reviewers' comments:

Reviewer's Responses to Questions

**Comments to the Author**

1. Is the manuscript technically sound, and do the data support the conclusions?

Reviewer #1: Yes

Reviewer #2: Yes

Reviewer #3: Yes

2. Has the statistical analysis been performed appropriately and rigorously? 

Reviewer #1: Yes

Reviewer #2: Yes

Reviewer #3: Yes

3. Have the authors made all data underlying the findings in their manuscript fully available?

Reviewer #1: Yes

Reviewer #2: Yes

Reviewer #3: Yes

4. Is the manuscript presented in an intelligible fashion and written in standard English?

Reviewer #1: Yes

Reviewer #2: Yes

Reviewer #3: Yes

5. Review Comments to the Author

Reviewer #1: The authors have addressed an interesting and relevant topic for the social decision-making literature. I found the authors to be very transparent in their procedures and I really liked that the authors made their data and scripts publicly available on OSF and that there is already a codebook etc. I have a few comments and suggestions that the authors should consider in a revision.

Introduction

1. In general, I like the introduction very much as it is written with a clear focus on the research topic. I have recently been involved in research that has shown that not only the receiver's emotions play a role in acceptance/rejection behavior, but also the proposer's emotions. Anger played a central role in this, see e.g., Weiß et al., 2020. In addition, in another paper we showed that the trait anger moderates behavior only in an adapted two-stage version of the UG, but not in the traditional version (Rodrigues et al., 2022). I think both articles are relevant for the authors' introduction.

Weiß, M., Rodrigues, J., Boschet, J. M., Pittig, A., Mussel, P., & Hewig, J. (2020). How depressive symptoms and fear of negative evaluation affect feedback evaluation in social decision-making. Journal of Affective Disorders Reports, 1, 100004.

Rodrigues, J., Weiß, M., Mussel, P., & Hewig, J. (2022). On second thought… the influence of a second stage in the ultimatum game on decision behavior, electro‐cortical correlates and their trait interrelation. Psychophysiology, 59(7), e14023.

2. P. 5, l. 85: Typo “Ultimatum-Gave”

3. The authors state that “the relationship and differences between the constructs irritability and anger have not been clearly established, and in this study the overlap between the two will be examined in a behavioral setting.” I was surprised that subsequently the hypotheses focused exclusively on irritability. If the authors want to empirically test the differences, the hypotheses should include this aspect of the research question in my opinion.

Methods

4. To my understanding, the authors did not use an a priori effect size to estimate significance, but a maximum number of participants they would be able to collect. How did you come to the assumption that your effect would be above the effect size of r = .19 when you decided to conduct the study? I didn't fully understand why the authors didn't collect more data until they reached their pre-registered sample size of 200 participants. With the online study, the time to recruit participants is drastically reduced.

5. P.12, l. 12: What was the “neutral filler task”?

6. How was the actual payout calculated for the participants? From reading the Method section, I was not sure whether the authors described a cover story/deception. I would encourage the authors to provide more details here.

7. I would suggest that the authors add an "Analysis" section to the "Methods" section so that the "Results" section focuses on the results.

Results

8. The described multiple regression model is not included in the script on OSF (“main_analysis.Rmd”)

9. Why did the authors not use a model to capture their actual research question? I would have assumed three linear models with the predictors anger, irritability, their interaction, group as covariate, and the two rejection rates (unfair, medium) and total gain in the AI-UG as the outcome variables, respectively.

The following R code yielded some interesting results for the total gain and the rejections for the medium offers.

#Read data into R

data <- haven::read_spss("UGdatafile.sav")

### Anger Trait

data$STAXI_trait_sum <- rowSums(data[which(colnames(data) == "STAXI_trait_1"):

which(colnames(data) == "STAXI_trait_10")], na.rm = FALSE)

### BITe_1 (first measure)

data$BITe_sum1 <- rowSums(select(data, "BITe_1","BITe_2","BITe_3",

"BITe_4","BITe_5"))

### BITe_2 (second measure)

data$BITe_sum2 <- rowSums(select(data, "BITe_1_2","BITe_2_2",

"BITe_3_2","BITe_4_2","BITe_5_2"))

# rejections

data$ug_m_rejection <- 420 -data$UG_Medium_sum

data$ug_u_rejection <-180 - data$UG_Unfair_sum

### Model for total gain

total_sum_model <- lm(UG_sum ~ scale(STAXI_trait_sum)*scale(BITe_sum1)*as.factor(Online_or_Lab)*scale(BITe_sum2)*as.factor(Online_or_Lab), data=data)

summary(total_sum_model)

sjPlot::plot_model(total_sum_model, type="pred", terms=c("STAXI_trait_sum", "BITe_sum1", "Online_or_Lab"))

### Model for unfair offers

unfair_model <- lm(ug_u_rejection~ scale(STAXI_trait_sum)*scale(BITe_sum1)*as.factor(Online_or_Lab)*scale(BITe_sum2)*as.factor(Online_or_Lab), data=data)

summary(ufair_model)

### Model for medium-sized offers

medium_model <- lm(ug_m_rejection ~ scale(STAXI_trait_sum)*scale(BITe_sum1)*as.factor(Online_or_Lab)*scale(BITe_sum2)*as.factor(Online_or_Lab), data=data)

summary(medium_model)

sjPlot::plot_model(medium_model, type="pred", terms=c("STAXI_trait_sum", "BITe_sum1", "Online_or_Lab"))

Please note that I may have misunderstood something here. I have only carried out the analysis as I understood your research question.

10. I don't want to encourage the authors to adjust their analyses post-hoc, but if they decide on a model, I would ask them to make it as transparent as possible (e.g., exploratory or deviation from pre-registered plan due to reviewer comments, etc.). I really like the study idea and would like to support the authors that they can get the most out of their data to answer their research question as precisely as possible.

11. If the authors decide for a model, I would suggest including figures on the main results.

Discussion

12. As the interpretation of the results might change according to my suggestion, I did not conduct an in-depth review of the Discussion.

Reviewer #2: I had the pleasure to revise the original manuscript entitled “An Evaluation of the Ultimatum Game as a Measure of Irritability and Anger”. The manuscript is clear and well-written. The behavioral results of this study showed significant relationships between anger and aggressive behaviors in the Ultimatum game, but no association between irritability and aggressive behavior as authors hypothesized.

I suggest addressing the following minor points:

-Page 5 – line 85: Gave should become Game.

-On page 12, line 264-265, refer to the administration of the PSAP. In what order was it administered relative to the UG? It is not insignificant that there was a task in the procedure that was later removed. From the initial instructions, did the participants know they had to perform two tasks?

-I suggest reporting the description of statistical analyses performed.

-Table 2, first line “KI” should be “CI”?

-page 17, line 357-364: Are these correlational results additional to those reported in table 1? If yes, please report all the statistical details of these results.

Reviewer #3: The paper is surely well structured and I'm sure the data analysis has been accomplished properly (this is why I marked the paper through the positive option at question 2 above) but it is not presented completely. Just to be ready for publication, I ask the author to insert at least a paragraph completing the description of the experimental design and data analysis with special mention to the type of statistical tests that have been used alongside their assumptions check.

6. PLOS authors have the option to publish the peer review history of their article (what does this mean?). If published, this will include your full peer review and any attached files.

Reviewer #1: No

Reviewer #2: **Yes: **Laura Angioletti

Reviewer #3: No

---

## [Author Response · Author response to Decision Letter 0]

15 Mar 2024

PONE-D-23-32978

An Evaluation of the Ultimatum Game as a Measure of Irritability and Anger

PLOS ONE

We’d like to thank you for the opportunity to resubmit a revised version of our manuscript. We are grateful for your and the reviewers’ careful reading of our original submission and thoughtful and constructive comments. We have done our best to accommodate as many of the suggested revisions as possible, and we feel that the paper has improved significantly as a result. Below, we summarize the revisions that we have made. 

The manuscript and files have been prepared according to the style requirements.

2. In the online submission form, you indicated that your data will be submitted to a repository upon acceptance. We strongly recommend all authors deposit their data before acceptance, as the process can be lengthy and hold up publication timelines. Please note that, though access restrictions are acceptable now, your entire minimal dataset will need to be made freely accessible if your manuscript is accepted for publication. This policy applies to all data except where public deposition would breach compliance with the protocol approved by your research ethics board. If you are unable to adhere to our open data policy, please kindly revise your statement to explain your reasoning and we will seek the editor's input on an exemption.

We have uploaded all relevant data files of the project, including data and analyses on OSF. During the review process, the material on OSF is anonymous (but available for the reviewers), but when accepted we will make the OSF page public. 

Comments to the Author

Reviewer #1: 

1. I have recently been involved in research that has shown that not only the receiver's emotions play a role in acceptance/rejection behavior, but also the proposer's emotions. Anger played a central role in this, see e.g., Weiß et al., 2020. In addition, in another paper we showed that the trait anger moderates behavior only in an adapted two-stage version of the UG, but not in the traditional version (Rodrigues et al., 2022). I think both articles are relevant for the authors' introduction.

We have read the two articles you suggested with great interest. The Rodriges et al. (2022) study fits nicely in the Introduction and has now been added (Line 101). While interesting and partially related to the current topic, we found it difficult to incorporate the Weiß et al. (2020) study without providing additional context, which would lead to a less streamlined Introduction section. 

2. P. 5, l. 85: Typo “Ultimatum-Gave”

This typo has been corrected in the manuscript.

3. The authors state that “the relationship and differences between the constructs irritability and anger have not been clearly established, and in this study the overlap between the two will be examined in a behavioral setting.” I was surprised that subsequently the hypotheses focused exclusively on irritability. If the authors want to empirically test the differences, the hypotheses should include this aspect of the research question in my opinion.

We have restructured the section The Current Study (lines 118 - 144). Now, the first part of the section describes our preregistered hypotheses, focusing on the relationship between irritability and AI-UG behavior. In the second part of the section, exploratory analyses are presented. The sentence 'the relationship and differences between the constructs irritability and anger have not been clearly established, and in this study, the overlap between the two will be examined in a behavioral setting' has been rewritten and is now located under the second exploratory part of the section. 

4. To my understanding, the authors did not use an a priori effect size to estimate significance, but a maximum number of participants they would be able to collect. How did you come to the assumption that your effect would be above the effect size of r = .19 when you decided to conduct the study? I didn't fully understand why the authors didn't collect more data until they reached their pre-registered sample size of 200 participants. With the online study, the time to recruit participants is drastically reduced.

We did not have an a priori assumption about a specific effect size before conducting the study. Instead, we made the judgment that the statistical power to detect a relationship of r = .19 or larger would render the study informative from both a practical and a theoretical standpoint. Thus, we considered it worthwhile to conduct the study with the sample of 200 participants that was within reach. Even with the slightly reduced power of our final sample, we consider the ability to detect a relationship of r = .23 or larger still to be informative and not far from the originally calculated power. We did not continue data collection until we had reached 200 participants, because the extreme-groups approach meant that we were limited to the highest and lowest quartiles of the BITe 1 scores. Despite repeated efforts, only 146 of the 200 participants in those groups agreed to participate in the second study.

5. P.12, l. 12: What was the “neutral filler task”?

We have added a brief description of the neutral filler task under the heading 'Procedure'. During the filler task, participants were instructed to locate and count all instances of the letter 'a' in a brief, neutral text, marking each one with a pencil. They were informed that the task assessed accuracy and concentration, and that it was not being timed. (Lines: 266-269).

6. How was the actual payout calculated for the participants? From reading the Method section, I was not sure whether the authors described a cover story/deception. I would encourage the authors to provide more details here.

We have added a sentence under the heading 'The Anger-Infused Ultimatum Game (AI-UG)' where we clarify that the motivational information provided was a cover story: "In reality, all participants received a payment equivalent to 10% of the total amount earned during the game." (Lines: 226 - 227).

7. I would suggest that the authors add an "Analysis" section to the "Methods" section so that the "Results" section focuses on the results.

We have added a section titled 'Data Analyses' under the Methods (lines 286 - 321). Additionally, we have removed the descriptive parts of data analyses from the Results section.

8. The described multiple regression model is not included in the script on OSF (“main_analysis.Rmd”)

The multiple regression models has been included in the script on OSF (“main_analysis.Rmd”).

9. Why did the authors not use a model to capture their actual research question? I would have assumed three linear models with the predictors anger, irritability, their interaction, group as covariate, and the two rejection rates (unfair, medium) and total gain in the AI-UG as the outcome variables, respectively.

10. I don't want to encourage the authors to adjust their analyses post-hoc, but if they decide on a model, I would ask them to make it as transparent as possible (e.g., exploratory or deviation from pre-registered plan due to reviewer comments, etc.). I really like the study idea and would like to support the authors that they can get the most out of their data to answer their research question as precisely as possible.

Thank you for your careful review of the manuscript's analyses and for your encouragement to explore possible models to analyze and understand our data most effectively and accurately. To stay consistent with our preregistration, we have chosen to retain the originally planned correlation analyses reported in Table 1. However, we have also considered additional analyses in line with your suggestions. We have taken a look at the R code you shared. While we found the three-way interaction detected by the model to be intriguing, it does not seem to be robust to subtle variations in analyses. For instance, when including anger expression and control as predictors in the model (which should be included to fulfill the exploratory purpose stated in the preregistration), the interaction effect ceases to be significant. Therefore, we are not confident enough that the effect is a “real” phenomenon to draw any conclusions from the interaction effect.

Thanks to your feedback, however, we have decided to report a series of hierarchical regression models where we include BITe 1 scores (Step 1), STAXI trait anger (Step 2), and STAXI anger control and expression subscales (Step 3) as predictors. Consistent with the correlational analyses, Steps 1 and 2 of this analysis shows that trait anger, but not irritability, significantly predicts participants’ gain in the AI-UG. (Lines: 383-291 and Table 3).

Reviewer #2: 

11. Page 5 – line 85: Gave should become Game.

This typo has been corrected in the manuscript.

12. On page 12, line 264-265, refer to the administration of the PSAP. In what order was it administered relative to the UG? It is not insignificant that there was a task in the procedure that was later removed. From the initial instructions, did the participants know they had to perform two tasks?

The test order was randomized (See lines 265 - 266) "To control for any order effects, the order of the games was randomized, and the two games were separated by a neutral filler task. 

Participants received information before coming to the lab that during the study, they would be answering questions and completing two tests in front of a computer. To further clarify in the Procedure section, we have added that participants received oral instructions "about the testing procedure". (Line 263).

13. I suggest reporting the description of statistical analyses performed.

We have added a section titled 'Data Analyses' under the Methods (lines 286 - 321). Additionally, we have removed the descriptive parts of data analyses from the Results section." 

14. Table 2, first line “KI” should be “CI”?

This typo has been corrected in the manuscript.

15. page 17, line 357-364: Are these correlational results additional to those reported in table 1? If yes, please report all the statistical details of these results.

The statistical details of the correlational results on page 17 are presented in Table 1. 

Reviewer #3: 

16. The paper is surely well structured and I'm sure the data analysis has been accomplished properly (this is why I marked the paper through the positive option at question 2 above) but it is not presented completely. Just to be ready for publication, I ask the author to insert at least a paragraph completing the description of the experimental design and data analysis with special mention to the type of statistical tests that have been used alongside their assumptions check.

We have added a section titled 'Data Analyses' under the Methods (lines 286 - 321). Additionally, we have removed the descriptive parts of data analyses from the Results section."

---

## [Decision Letter · Decision Letter 1]

6 May 2024

An Evaluation of the Ultimatum Game as a Measure of Irritability and Anger

PONE-D-23-32978R1

Dear Dr. Maria Gröndal,

We’re pleased to inform you that your manuscript has been judged scientifically suitable for publication and will be formally accepted for publication once it meets all outstanding technical requirements.

Kind regards,

Junchen Shang

Academic Editor

PLOS ONE

Additional Editor Comments (optional):

Reviewers' comments:

Reviewer's Responses to Questions

**Comments to the Author**

1. If the authors have adequately addressed your comments raised in a previous round of review and you feel that this manuscript is now acceptable for publication, you may indicate that here to bypass the “Comments to the Author” section, enter your conflict of interest statement in the “Confidential to Editor” section, and submit your "Accept" recommendation.

Reviewer #1: All comments have been addressed

Reviewer #2: All comments have been addressed

Reviewer #3: All comments have been addressed

2. Is the manuscript technically sound, and do the data support the conclusions?

Reviewer #1: Yes

Reviewer #2: Yes

Reviewer #3: Yes

3. Has the statistical analysis been performed appropriately and rigorously? 

Reviewer #1: Yes

Reviewer #2: Yes

Reviewer #3: Yes

4. Have the authors made all data underlying the findings in their manuscript fully available?

Reviewer #1: Yes

Reviewer #2: Yes

Reviewer #3: Yes

5. Is the manuscript presented in an intelligible fashion and written in standard English?

Reviewer #1: Yes

Reviewer #2: Yes

Reviewer #3: Yes

6. Review Comments to the Author

Reviewer #1: The authors have addressed all my comments to my satisfaction and I recommend the work for publication.

Reviewer #2: (No Response)

Reviewer #3: The Authors addressed all comments of previous reviewers, in my opinion, and I have nothing to add.

The paper can be accepted in its current form.

7. PLOS authors have the option to publish the peer review history of their article (what does this mean?). If published, this will include your full peer review and any attached files.

Reviewer #1: No

Reviewer #2: No

Reviewer #3: No

---

## [Editor Report · Acceptance letter]

14 Jun 2024

PONE-D-23-32978R1 

PLOS ONE

Dear Dr. Gröndal, 

I'm pleased to inform you that your manuscript has been deemed suitable for publication in PLOS ONE. Congratulations! Your manuscript is now being handed over to our production team.

Kind regards, 

on behalf of

Dr. Junchen Shang 

Academic Editor

PLOS ONE